# Direct Energy Deposition Parametric Simulation Investigation in Gear Repair Applications

**DOI:** 10.3390/ma16093549

**Published:** 2023-05-05

**Authors:** Nuno Miguel Ferreira, Maria Vila Pouca, Carlos Fernandes, Jorge Seabra, Grzegorz Lesiuk, Marco Parente, Abílio Jesus

**Affiliations:** 1DEMec—Department of Mechanical Engineering, Faculty of Engineering, University of Porto (FEUP), Rua Dr. Roberto Frias, 4200-465 Porto, Portugal; nmferreira@fe.up.pt (N.M.F.); up201202768@edu.fe.up.pt (M.V.P.);; 2INEGI—Institute of Science and Innovation in Mechanical and Industrial Engineering, Faculty of Engineering, University of Porto, Rua Dr. Roberto Frias 400, 4200-465 Porto, Portugal; 3Department of Mechanics Materials Science and Biomedical Engineering, Faculty of Mechanical Engineering, Wroclaw University of Science and Technology, PL 50370 Wroclaw, Poland

**Keywords:** metal additive manufacturing, finite element method, DED, gear repair

## Abstract

Additive manufacturing technologies have numerous advantages over conventional technologies; nevertheless, their production process can lead to high residual stresses and distortions in the produced parts. The use of numerical simulation models is presented as a solution to predict the deformations and residual stresses resulting from the printing process. This study aimed to predict the tensions and distortions imposed in the gear repair process by directed energy deposition (DED). First, the case study proposed by National Institute of Standards and Technology (NIST) was analyzed to validate the model and the numerically obtained results. Subsequently, a parametric study of the influence of some of the parameters of DED technology was carried out. The results obtained for the validation of the NIST benchmark bridge model were in agreement with the results obtained experimentally. In turn, the results obtained from the parametric study were almost always in line with what is theoretically expected; however, some results were not very clear and consistent. The results obtained help to clarify the influence of certain printing parameters. The proposed model allowed accounting for the effect of residual stresses in calculating the stresses resulting from gear loading, which are essential data for fatigue analysis. Modeling and simulating a deposition process can be challenging due to several factors, including calibrating the model, managing the computational cost, accounting for boundary conditions, and accurately representing material properties. This paper aimed to carefully address these parameters in two case studies, towards reliable simulations.

## 1. Introduction

Additive manufacturing has gained increased relevance in recent years. This unconventional manufacturing process is an important complement to the existing traditional processes [1]. Its appearance has changed the way engineering design is seen, as it allows the production of components that would be impossible to produce using traditional manufacturing processes [2].

Within this new universe that is additive manufacturing, metal processing is currently at the forefront of the development field [3,4]. Due to the evolution of the process and continuous improvements, these techniques are now capable of producing high-strength parts with good mechanical characteristics [5,6]. On the other hand, some challenges still need to be overcome such as the high residual stresses produced, which can have a significant impact on fatigue characteristics and cause geometric distortions [7].

For cost-effective fabrication of end-use metallic AM parts, three processes are commonly used: selective powder sintering, selective powder melting, and DED (with powder or wire feeding). The primary heat sources for those processes include lasers, electron beams, electric arcs, and plasma [8].

Powder bed fusion (PBF) processes were among the first commercialized AM processes. All PBF processes share a basic set of characteristics. These include one or more thermal source for inducing fusion between the powder particles, a method for limiting the powder fusion to a prescribed region of each layer, and mechanisms for adding and smoothing the powder layers [9,10].

In the powder bed fusion process, thin layers of powder are applied to a build plate, and an energy source (a laser or electron beam) is used to fuse the powder at the locations specified by the model of the desired geometry. When one layer is completed, a new layer of powder is applied, and the process is repeated until a three-dimensional part is produced. The PBF process is alternatively known as selective laser sintering (SLS), selective laser melting (SLM), directed metal laser sintering (DMLS), directed metal laser melting (DMLM), and electron beam melting (EBM) [11].

Unlike powder bed fusion techniques, DED processes are not used to melt a material that is prelaid in a powder bed but are used to melt materials as they are being deposited [9]. One of the most successful applications of DED is in repairing metal parts by adding material to damaged components, such as turbine blades.

A number of organizations have developed DED machines using lasers and powder feeders. These machines are referred to as laser engineered net shaping (LENS), directed light fabrication (DLF), directed metal deposition (DMD), 3D laser cladding, laser generation, laser-based metal deposition (LBMD), laser freeform fabrication (LFF), laser directed casting, LaserCast, laser consolidation, LasForm, and others [12]. Although the general approach is the same, differences between these machines commonly include changes in laser power, laser spot size, laser type, powder delivery method, inert gas delivery method, feedback control scheme, and/or the type of motion control utilized [9].

The DED and PBF processes are the predominant technologies used worldwide; therefore, several articles have been published to verify the applicability of DED for repairing damaged parts in various industries [13,14,15], or even for using DED to repair parts produced in PBF [16].

The use of numerical models is presented as a solution that allows accurately and efficiently predicting the behavior of parts [17,18,19,20]. This study used numerical simulation software to study certain additive metal manufacturing technologies, namely the powder bed fusion process and the directed energy deposition process. Abaqus was the software used to simulate these processes. Abaqus has an additive manufacturing (AM) module, an interface that allows the user to prepare complex additive manufacturing simulations [21]. The results of predicting distortions and residual stresses were achieved through an uncoupled thermomechanical finite elements analysis [22].

Since it was not possible to validate the gear numerical model with experimental data, the efficiency of the numerical model was validated using the results obtained from the NIST AMB2018-01 Benchmark study [23]. A comparison was made between the numerical and experimental data related to residual stresses and the deflection of the benchmark bridge when gradually removing the part from the substrate, using the simulation finite element method, allowing us to determine the accuracy of the developed model [24].

This parametric study was motivated by the GEAR3D project, which consists of the 3D reconstruction of large gear teeth for the wind power sector, via direct laser deposition, ensuring mechanical properties equal to or greater than those of the new components [25]. The parametric study was carried out to evaluate the influence on the numerical results of residual stresses and distortion for a given gear specimen. A reduced-scale gear was used to carry out this study because carrying out a full-scale study of these significant mechanical components would lead to a very high cost.

Some works have already focused on this problem, and an advanced automated damage detection and damage reconstruction algorithm for damaged gear tooth repair has already been proposed [26].

This work focuses on the parametric investigation of the DED technology used to reconstruct mechanical components, namely gears, aiming to demonstrate the possibility of predicting the residual stresses resulting from an additive manufacturing process.

## 2. Problem Definition

The objective of this paper was the numerical simulation of the reconstruction of mechanical components using a directed energy deposition additive manufacturing process. Before simulating the repair of the gear through the DED process, a preliminary simulation was carried out to validate the numerical procedures. The AMB2018-01 Benchmark bridge simulation from NIST [23] was analyzed, to allow validating the model created and the results obtained. This case study used PBF technology, which was expected to simulate the process of production, cooling, and structural loading of the bridge.

It was intended in this work to conduct a thermostructural study of gear repair with DED, using the finite element method via Abaqus, to understand the origins of residual stresses and the impact of different process parameters. For this parametric study, the following simulations were performed:Mesh: Convergence studies to choose a mesh that allows obtaining accurate results with the lowest possible computational cost under user-defined parameters;Step Time: The time step has a direct influence on the number of elements that are activated, and therefore it is also an important parameter to be studied;Preheat temperature: Study of the influence of different preheat temperatures;Scanning strategies: Comparison of different laser scanning strategies, to study the impact of this parameter on the residual stresses;Material: The use of different materials has a direct influence on the residual stresses produced, hence the importance of their study.

### 2.1. Benchmark Bridge Simulation

The AMB2018-01 tests consist of laser powder bed fusion (LPBF) 3D metal alloy builds of a bridge structure geometry that has 12 legs of varying sizes. The 12 legs consist of four replications of the section described in the green box in Figure 1. The primary objectives of the AMB2018-01 tests are to investigate residual stress within the structure, the part distortion that occurs after a section of the part is cut via wire electron discharge machining (EDM) [23].

As shown in Figure 2, the bridge structure is 75 mm long, 5 mm wide, and 12 mm tall, with with twelve legs that are of three different sizes—5 mm, 0.5 mm, and 2.5 mm, respectively. The thickness of 0.5 mm requires a refined mesh for reasonable results. Additionally, the bridge has eleven prominences at the top with 1 × 5 × 0.5 mm, and with a spacing between them of 6 mm.

The bridge is positioned on a build plate. The build plate structure is 90 mm long, 15 mm wide, and 12.7 mm tall.

### 2.2. Gear Repair

The DED process can be used to repair various mechanical components, including gears. Gears are a very useful transmission mechanism that is used to transmit rotation from one axis to another. The gear system can be used to decrease the speed (and also increase the torque). Gears are commonly used in high-load situations, because the teeth of the gear allow finer, more discreet control of the movement of a shaft. Due to the high load situations to which gears are subjected, it is important to ensure that repairs are capable of withstanding these forces.

To simulate the repair of a damaged gear, two main elements were considered, the tooth and the gear with a cut tooth, as shown in Figure 3, kindly provided by INEGI [27]. In this figure, it is possible to observe the three stages of the gear rebuilding process. In this work, only the residual stresses caused by the DED process were considered, the residual stresses resulting from the machining, Figure 3c, may be studied in future work and would result in a smaller material depth from the surface (∼0.1 mm).

The gear in Figure 3 was repaired at INEGI, using a KUKA KR 30 HA robot equipped with a Fraunhofer COAX12V6 laser cladding head, as shown in Figure 4.

The parameters used in the DED process to repair the gear in Figure 3 are shown in Table 1. The repair of this gear was carried out at INEGI and served as a starting point for the parametric study carried out.

The models for numerical simulation were created from the technical drawing of the pinion, Figure 5. This pinion is type C and has 16 teeth and an internal diameter of 30 mm, the remaining dimensions are indicated in Figure 5.

### 2.3. Material Definition

The material for the benchmark bridge simulation was nickel-based superalloy IN625. In the parametric study of the DED process, IN625 was always used. In addition to this material, 42CrMo4 and 18Ni300 Maraging are used only in numerical simulation in order to study the influence of different materials.

#### 2.3.1. Inconel 625

IN625 is a nickel-based superalloy known for its high level of strength, temperature resistance, and corrosion resistance. The strength of IN625 is derived from the stiffening effect of molybdenum and niobium in its nickel–chromium matrix, thus precipitation–hardening treatments are not required [28].

While conventional manufacturing of components using these high-performance alloys has been difficult with machining, due to excessive tool wear and low material removal rates, powder-based AM technologies can remove these constraints, while improving lead times and reducing manufacturing costs [29].

Latent heat effects can be significant and must be included in heat transfer problems involving a phase change. The latent heat is assumed to be released over a range of temperatures, from a lower (solidus) temperature to an upper (liquidus) temperature, Table 2. To model a pure material with a single phase change temperature, these limits can be made very close.

The latent heat is the extra energy required to achieve the phase change in the material, in the form of thermal energy absorption (melting) or release (solidification). Latent heat can be combined with any other material behavior in Abaqus, but it should not be included in the material definition unless necessary; it adds an extreme non-linearity to the thermal solution, which may inhibit convergence.

The temperature dependence properties can be seen in Table 3. In order to couple the thermal with the mechanical simulation, the temperature field result was introduced as a time-dependent boundary condition for the static solution. A gradient of temperature causes a deformation in the solid, which linearly depends on the expansion coefficient of the material. The material density was 8.44 $\times 10^{-9}$ (ton/mm${}^{3}$).

Young’s modulus is a mechanical property of solids, which gives the stiffness of the material. It can also be defined as the ability of the material to withstand changes in its shape under expansion or compression. The Young’s modulus of a material is defined as the ratio of longitudinal stress to the linear strain; as the temperature rises, Young’s modulus decreases, Table 4.

#### 2.3.2. 18Ni300 Maraging Steel

The term “maraging” is a portmanteau of martensite and aging and represents the age hardening of a low-carbon, iron–nickel martensitic matrix. Maraging steel is a class of high-alloy, low-carbon steel developed for structural applications requiring high strength. This family of alloys have good weldability, due to the practically complete absence of interstitial alloying components. As a result, they are conducive to metal additive manufacturing procedures such as laser metal deposition (LMD) and selective laser melting (SLM) [30].

The qualities of this material have not been well established and documented; as a result, the thermal property values were obtained experimentally [31] and are shown in Table 5. The Poisson’s ratio and Young’s modulus values were 0.3 and 190 GPa, respectively, and the density was 8 $\times 10^{-9}$ ton/mm${}^{3}$. In terms of the material’s plastic properties, a J-C model was used with values from the literature, which are shown in Table 6.

The Johnson–Cook constitutive model is not based on traditional plasticity theory and reproduces several important material responses observed in the impact and penetration of metals. Strain hardening, strain-rate effects, and thermal softening are the three main material responses. The Johnson–Cook constitutive model multiplicatively combines these three effects [33].
(1)$$\sigma =\left(A+(B\varepsilon^{n})\right)\left(1+C\ln\left(\frac{\dot{\varepsilon }}{{\dot{\varepsilon }}_{0}}\right)\right)\left(1-(T_{h}^{m})\right)$$
where ε is the plastic strain, $\dot{\varepsilon }$ is the plastic strain rate, $\dot{\varepsilon_{0}}$ is the reference plastic strain rate, *A* is the static yield strength, *B* is the strain hardening coefficient, *n* is the strain hardening exponent, *m* is the thermal sensitivity parameter, and $T_{h}$ a nondimensional homologous temperature defined as: (2)$$T_{h}={\left(\frac{\left(T-T_{ref}\right)}{\left(T_{m}-T_{ref}\right)}\right)}^{m}$$

In Equation (Equation 1), the first bracketed term represents the strain hardening of the yield stress, the next term models the increase in the yield stress at elevated strain rates, and the final bracketed term is a softening of the yield stress due to local thermal effects.

Table 6 shows the values obtained in the literature [20] for the plastic properties using the Johnson–Cook model.

#### 2.3.3. 42CrMo4 Steel

42CrMo4 (1.7225) steel is a low-alloy steel that contains chromium, molybdenum, and manganese. The presence of chromium and molybdenum increases the corrosion resistance. Molybdenum is especially useful in resisting the corrosion caused by chlorides. This steel has different grades: the European designation is 42CrMo4 (1.7225), and the USA grade is 4140.

This steel is widely used in a variety of industries such as aerospace, oil, and gas, as well as many other industries. The thermophysical properties of the material 42CrMo4 are given in Table 7, and the density is 7.8 $\times 10^{-9}$ (ton/mm${}^{3}$).

To characterize the plastic regime of this material, the Johnson–Cook constitutive model was used—Equations (Equation 1) and (Equation 2). In Table 8 are the values found in the literature, [35] and [36], respectively.

The latent heat properties are shown in Table 9.

## 3. Numerical Details

In a powder-bed-type additive manufacturing method, a recoater or a roller blade deposits a single layer of raw material. The part is then scanned in a single cross-section with a high-powered laser across the raw material layer to fuse it with the previously deposited layer beneath. Layer-upon-layer raw material deposition is represented by a progressive element activation in structural or thermal analysis, while laser-induced heating is represented by a moving heat flux [37].

During the laser-directed energy deposition (LDED) method, the material is deposited by a nozzle positioned on a multi-axis arm, while being melted by a laser beam. In structural analysis, a beginning temperature denoting a relaxation temperature above which thermal straining causes low thermal stress is typically assigned, to accurately capture the melting effect [38].

### 3.1. Benchmark Bridge

#### 3.1.1. Modeling

In the additive manufacturing benchmark test simulation, two parts were used— Figure 6. The bridge was positioned in the center of a build plate.

Several simulations were performed, until the model was fine-tuned, to give consistent results. IN625 was used in both parts, and the results obtained from this simulation were later compared with the experimental results presented by NIST.

#### 3.1.2. Initial Conditions

In the heat transfer analysis, an initial temperature of 80 °C was applied to the build plate. The initial temperature of the part was 40 °C, which was the temperature at which the powder material was dispensed from the powder bed reservoir.

In structural analysis, the initial temperature of an assembly represents the relaxation temperature above which thermal strain results in negligible thermal stress. In material activation, it represents the temperature at which the initial thermal shrinkage occurs [21].

In this analysis, the initial temperature of the bridge was set to 750 °C. The initial temperature of the build plate was 80 °C.

#### 3.1.3. Mesh—Bridge

A 3D 8-node linear brick element was employed in both the thermal and mechanical simulations. The structural element had the designation C3D8 and the heat transfer element was DC3D8. Along the printed area, the spatial distribution of the elements was uniform or “mapped.” The printing simulation results were influenced by the element sizes.

The first letter of the element’s name indicates the family to which the element belongs, in this case, the C in C3D8 indicates this is a continuum element. The number 3 represents the degrees of freedom.

The mesh of the bridge structure consisted of 8-node linear diffusive heat transfer DC3D8 elements. The elements varied in size but had a common characteristic height of 0.2 mm, so that there were approximately ten layers of material in each element. The build plate was more coarsely meshed with DC3D8 elements. In the static analysis, C3D8 solid elements were used for both the bridge and the build plate, maintaining the same mesh topology used in the heat transfer analysis.

#### 3.1.4. Progressive Element Activation

A laser beam scans at a controlled speed the selected locations of the powder bed and fuses the powder, then the powder bed is lowered by a defined layer thickness and a new layer of powder is added. Progressive element activation can be used in Abaqus to represent layer-upon-layer raw material deposition in a thermal-stress analysis.

A sequential thermal stress analysis of an additive manufacturing process consists of a transient heat transfer analysis of the thermal loads introduced by the process, followed by a static structural analysis that is driven by the temperature field from the thermal analysis [39].

During the analysis, it is possible to control the activation and the volume fraction of the material for each element in each increment. Any element that is defined as progressively activated is filled with material or remains empty (inactive).

To define the material deposition process, it is necessary to consider the following steps [40]:Heating source: The toolpath–mesh intersection module takes the time–location history and automatically computes all of the information required to activate elements and apply the proper thermal energy to the model.Recoater motion: The infinite line representation is useful for describing this process of layer-by-layer material deposition. All path segments when the tool is in the “on” state must be perpendicular to the global z-direction.Table collections: Table collections that encapsulate parameter tables or property tables can be used to define additional process parameters needed for the simulation.

#### 3.1.5. Scanning Strategy

The scanning strategy is the spatial motion pattern of the energy beam (laser, electron beam, electrical arc, and so on). For a single-layer scan, the scanning strategy varies with different scanning directions, scanning sequences, scanning vector rotation angles, scanning vector lengths, scanning times, and hatch spaces [41].

The impact of the scanning strategy during the processing of different materials has been the subject of numerous research works. The laser speed, the solidification direction imposed by the moving heat source, the hatch space, and the cooling rate are some of the parameters that greatly influence the results obtained. These parameters influence the mechanical properties obtained, as well as the residual stress, porosity levels, and crystallographic texture of the material [42].

For this simulation, a scan strategy was used in which for odd-numbered layers, the infill scans were horizontal lines (parallel to the X-axis) that were separated by 0.1 mm (hatch spacing), and for even-numbered layers, the infill scans were vertical lines (parallel to the Y-axis) that were also separated by 0.1 mm. Figure 7 shows the scan pattern strategy description.

The process starts with the laser going through a full layer of the bridge, then the laser is disabled, and the roller is activated. The roller extends a thin layer of metallic powder, and then the laser is activated again to go through another layer. This process is repeated until the final structure of the bridge is obtained. Each powder layer is 0.02 mm thick, and a total of 625 layers are needed to build the bridge structure. In Figure 8 can be seen the slicing carried out to obtain the necessary event series for the simulations.

Table 10 shows the main parameters used to generate the path traveled by the laser.

#### 3.1.6. Cooling

The two heat-transfer mechanisms of cooling to the environment are convection and radiation. Both were subject to a sink temperature of 25 °C. Except for the bottom surface of the build plate, all of the model’s free surfaces were set to a typical film coefficient for an inert gas atmosphere of 0.018 mW/(mm${}^{2}$ °C). The emissivity of the exposed surface to the atmosphere is a significant property of radiation, and the chosen value was 0.45.

#### 3.1.7. Postprocessing Simulation

Additional postprocessing activities for a printed item, such as wire electrical discharge machining (EDM) to remove a part from a build plate or to remove support structures, heat treatment, or other subsequent machining processes, are frequently required to be simulated [40].

After the build was completed, the parts were cut using wire electrical discharge machining, so that only the ends of the parts remained attached to the plate. In the simulation, the cutting process with wire EDM was modeled using progressive removal of specified elements in the cutting region. This was modeled in a separate step using a model modification to remove a layer of elements near the bottom of the legs. One of the ends of the part was still attached to a portion of the build plate. The cut part of the bridge bent upward, due to the relaxation of residual stress, Figure 9.

### 3.2. Gear Simulation

The GEAR3D project [25] was the starting point for the simulations carried out in this chapter. This project aimed to repair damaged mechanical components, namely wind turbine gears. Additive manufacturing was used to repair these gears, more specifically the DED process. The parameters used in the simulations were provided by INEGI [27].

The main objective was to perform a parametric study evaluating the influence of the various parameters of the simulation. Thus, different materials, mesh sizes, scanning strategies, preheat temperatures, and step times were the five parameters studied.

#### 3.2.1. Modeling

During the process of modeling the gear, several attempts were made to obtain a reliable model. One of the first possibilities was to model two pieces separately. One of the pieces would be the substrate—the gear without the tooth that would be rebuilt, and the other piece would be the shape of the tooth to be rebuilt by DED. Both parts are represented in Figure 10a.

Another possibility was the design of a single piece, Figure 10b. In order to simulate the tooth reconstruction, all elements were deactivated. The elements would be activated later, when the part was DED printed. Modeling two separate parts leads to different meshes on the parts. The simulation of two parts with different meshes can lead to the problem of discontinuity between meshes.

The mesh discontinuity problem becomes evident through the detachment of the two pieces when the obtained results are analyzed. Thus, only one of the mentioned possibilities could be considered for the simulations.

#### 3.2.2. Mesh

For the study of this parameter, three different meshes were used. The most discretized mesh had an approximate size of 0.25 mm and will be named Mesh-1, the second mesh had an approximate size of 0.5 mm and will be referred to as Mesh-2, finally, the coarsest mesh had an approximate size of 1.5 mm and will be referred to as Mesh-3. In Figure 11, the three meshes used for these simulations can be observed. Discretization was applied to the whole gear model, because the printing process simulation does not guarantee symmetrical thermal loading.

The material used in the mesh study was IN625, the same for the entire gear. The strategy chosen was alternate hatches—single pass.

The mesh sizes presented above, and the number of elements related to each mesh are organized in Table 11. The number of elements for each mesh is represented in Figure 12. In this way, the difference between the number of elements of each mesh becomes more noticeable.

#### 3.2.3. Scan Strategy

The literature describes various methods for mitigating the residual stress effects of residual stress on a printed part [41,42,43]. One of the most effective ways to reduce the impact of residual stress is to manipulate the scanning strategy. Scanning strategy refers to the manipulation of laser specifications, such as the laser power, scanning speed, and the laser scanning pattern.

For this parametric study, the strategies given in Figure 13 were used. These scanning strategies were the zig-zag type, and angles of variation of 120°, 60°, and 0° were used.

The scanning strategy used here was zig-zag, and the final piece was obtained by deposition of layers with different angles of deposition. Before the filling material is deposited, a layer is deposited on the edge.

#### 3.2.4. Preheat Temperature

Preheating involves raising the temperature of the base metal above the ambient temperature, before the DED process begins. By preheating the base plate, the thermal gradients are lowered and stresses can be reduced, avoiding the cracks and delamination resulting from residual stresses [44].

During the initial depositions of the AM process, reduction of thermal stresses is critical, due to the presence of the substrate base, which acts as a heat sink and thus causes the formation of even higher thermal gradients and cooling rates [45].

Table 12 describes the preheating temperature values used in the study of this parameter. Figure 14 on the right graphically shows the variation in preheating temperature.

#### 3.2.5. Time Step Influence

Assume that two analyses are performed to simulate two different time increments: a small time increment activating one element per increment, and a large time increment activating two elements per increment. In the two stress analyses, the initial configuration of every second element differs, resulting in different residual stresses and distortions [21].

Table 13 shows the variation of the step time parameters, the initial increment, and the maximum increment.

#### 3.2.6. Material

The objective of this work was to make a parametric study of some of the important variables in the repair of gears. Therefore, it was important to consider the use of different materials when repairing gears, Figure 15. So far in the simulations carried out, the same material (IN625) was considered in the substrate and part. However, this does not happen in the industry, because IN625 is too expensive to be used in large gears. Normally, this type of gear is built with more economical materials, such as 42CrMo4 steel, and later receives a thermal treatment to acquire an adequate surface hardness.

Thus, to study the influence of the material, it was considered that the base plate (the gear) was built from 42CrMo4 steel and the tooth printed by DED was made of IN625.

## 4. Results and Discussion

### 4.1. Benchmark Bridge

For simulation of the AM process, Abaqus offers two options for additive process simulation, which allow for the definition of appropriate boundary conditions, loads, interactions, constraints, and temperature-dependent material properties.

In the simulations performed, an uncoupled thermomechanical simulation was used, which allowed exact specification in time and space of the processing conditions.

#### 4.1.1. Elastic Strain

The results of the simulation are shown in Figure 16a and Figure 17a. Figure 16 shows a contour plot of the residual strain simulation results and experimental measure results in the x-direction. Similarly, Figure 17 shows the simulation and experimental results of the residual strain in the z-direction.

The x strain components were mostly tensile, as shown in Figure 16. The highest tensile values were found near the top and bottom of the sample. When the sample was released from the substrate, the tensile region concentrated at the top of the sample caused a bending moment that drove the distortion of the sample.

When the simulation results were compared to the NIST results, a few key features show significant agreement between the simulation and experimental results. A comparison of the simulation and experimental contour plots, Figure 16, in the x direction shows that both results indicated tensile strains across the main body of the part, with compressive strains on the left side of each leg, as well as at the ends. Besides this, the magnitude was of a very similar scale.

In a similar manner, the comparison of contour plots in the z-direction revealed a very high level of convergence. Figure 17, shows that the Z strain component of the residual strain was compressive through the sample’s interior, while the sample’s sides exhibited high-tensile regions. Again, the magnitudes were very close, with a peak value of $3.95\times 10^{-3}$ in the simulations and $3.5\times 10^{-3}$ in the experimental data.

#### 4.1.2. Residual Strain Comparison between XRD, ND, and Simulation Results

As both neutron diffraction—ND and X-ray diffraction—XRD techniques measured lattice spacings, it was preferable to compare residual strains rather than calculated stresses. This was especially true because the synchrotron X-ray method could not measure all three orthogonal components required to calculate stresses reliably. Only strains in the X and Z directions could be directly compared using the neutron and X-ray results [46].

Figure 18 shows the line profile for the XRD measurements (orange line), ND measurements (gray line), and simulation results (blue line) along the longitudinal (X direction) of the sample.

It should be noted that the XRD measurements and simulation results were taken from X = 0 mm to X = 75 mm, and the ND measurements were taken from X = 0 mm to X = 60 mm.

Figure 18 compares the strain results in the vertical (build) direction. Near the left edge of the sample (X = 0 mm), ND shows almost zero vertical strain, while the XRD and simulation results consistently show a drastic change in strain, from highly tensile (positive) to compressive (negative) from the left edge of the sample.

However, the results near the edges of the sample once again reflect ND’s inability to capture large strain gradients.

On the other hand, the results obtained from the numerical simulation (blue line) were almost always slightly superior to those obtained from measurements.

#### 4.1.3. Deflection Analysis

After being built, the parts remained on the build plate. The tops of the 11 ridges were skim cut, to remove the rough as-built surface. The vertical height, relative to the base plate, of the top of both side edges of each ridge was measured. The upward deflection results are presented in Figure 19, blue curve. The defection measurements were all positive, indicating that the part was deformed upward after being separated from the build plate using EDM.

In Figure 20, it is possible to see the results obtained from the Abaqus simulation regarding the deflection of the bridge. As already shown in the graph in Figure 19, the deflection was maximum at the left end of the bridge and decreased until it was very close to zero at the opposite end.

The experimental values of deflection achieved by NIST, and measured with sensors [47,48], are very similar to the numerical results, although at the right end of the bridge main support there is a small divergence.

### 4.2. Gear Simulation

In this section, the results obtained from the parametric study performed are reported. The residual stresses presented in this section correspond to the Von Mises stresses.

#### 4.2.1. Mesh Size

This section investigates the effects of mesh size on the residual stress. For the three different sizes shown in Figure 11, three simulations were run using the same scanning strategy. The Figure 21 and Figure 22 show the output results for each variable.

The results obtained for the three meshes studied are grouped in Table 14. In Figure 23 and Figure 24 can be seen graphically the results presented in Table 14.

Analyzing Figure 21, it is noticeable that the results obtained for meshes 1 and 2 are very similar. On the other hand, the residual stress of mesh 3 is noticeably different when compared to the other meshes. Given its mesh size of 1.5 mm, mesh 3 may not have provided an accurate description of the residual stresses developed in the gear during the AM process.

Computational cost is a parameter that must be taken into account, because one of the objectives of this type of numerical simulation is to obtain results in the shortest possible time. This compromise between computational cost and accurate results must be balanced, in order to have credible results in an acceptable time frame.

In this way, Figure 22 makes the results related to computational time already presented in Table 14 more perceptible.

After analyzing the results it becomes clear that Mesh 2 is the best option. This mesh presented results very similar to mesh 1; however, the computational cost was much lower, due to the lower refinement of this mesh. Although mesh 3 had the lowest computational cost, the results obtained do not suggest that it presented an adequate description of the process.

Therefore, mesh 2 was used in the subsequent numerical simulations.

Figure 23 and Figure 24 show the results of the simulations of the influence of the mesh size for residual stresses and spatial displacement, respectively, for different mesh options. Concerning the residual stresses, it is clear that Figure 23c has the highest residual stress value and this was located at the root of the tooth.

Figure 24 shows the results of the simulations of the influence of the mesh size on the spatial displacement. As already shown in Figure 21b, meshes 1 and 2 had very similar values; on the other hand, the results of mesh 3 were much lower.

#### 4.2.2. Scanning Strategy

The scanning strategy is a key factor in determining the thermal history and melt channel combination within a part in a laser powder bed fusion (LPBF) process, which can significantly affect the microstructural evolution, defect formation, and mechanical properties of the part [49].

There are numerous ways to scan a single layer, and various scan strategies have been investigated in terms of residual stresses. In this section, the influence of the three types of scanning [0°, 60°, and 120°] already presented in Section 3.2.3 will be studied.

In Figure 25, the residual stress and spatial displacement results obtained for the three scanning strategies are graphically represented. Table 15 also shows the numerical results obtained for each of the scanning strategies.

Analyzing the results, it was concluded that both strategies 1 and 2 presented very similar results, with a negligible variation. In turn, the third scanning strategy presented values lower than those obtained by the previous strategies.

Figure 26 and Figure 27 show the graphical results obtained for residual stress and spatial displacement, respectively.

Again, residual stresses were concentrated at the root of the tooth; however, with scanning strategies 1 and 2, they propagated along the tooth, as shown in Figure 26a,b.

#### 4.2.3. Preheat Temperature

The preheating temperature has a lot of influence on residual stresses and distortion. It was expected that by increasing the preheat temperature in the base plate (gear), the residual stresses and warping of the printed part (tooth) would decrease.

The numerical results obtained for the preheat temperature variation are summarized in Table 16.

The results obtained are shown in Figure 28 and prove what was expected. With the increase in the preheating temperature, there was a decrease in the residual stresses of the gear. This reduction was significant, since there was a reduction of 14% between the minimum preheat temperature of 0 degrees and the maximum preheat temperature analyzed.

Analyzing the obtained spatial displacement results, a large increase is visible, as the preheat temperature also increased. Comparing the results obtained, an increase of 527% from the minimum preheating temperature to the maximum temperature was verified.

The images taken from Abaqus and grouped in Figure 29 and Figure 30 illustrate the wide range of results obtained. For low preheat temperatures, the spatial displacement was low (blue mark); on the other hand, for high preheat temperatures, there was a large spatial displacement (red mark).

#### 4.2.4. Time Step Influence

The results obtained from the step time variation are presented in Table 17 and in Figure 31. The study of this parameter aimed to determine the influence of step time on the results obtained in the simulation.

The time step is a parameter that influences the activation of the elements, so if the chosen step time is higher, more elements will be active at once, which reduces the quality of the process simulation and consequently decreases the quality of the results obtained. Therefore, it was expected that the simulations with high time step values would present less accurate results; conversely, simulations with lower time steps would present more accurate results.

Five simulations were performed, varying only the initial increment and maximum increment, and the total step time was 60. Analyzing the results obtained, see Figure 31, it is possible to verify that, despite the significant variation of the parameters, there was no significant difference in the results. The spatial displacement variation was approximately 8.4%; however, the residual stress variation was less than 0.2%.

Note that the graphs in Figure 31 use a custom scale, to be able to see the small differences between the simulations.

Figure 32 shows a graphic evolution of the simulations obtained through the variation of the time step. As can be seen, there was no significant difference.

In turn, in Figure 33 is a graphic evolution of the spatial displacement variation. Here, there are no significant changes either, there is only a small decrease in the red spot (indicating higher spatial displacement values) as the time step decreases.

#### 4.2.5. Material

In this section, two different materials were used for tooth reconstruction; IN625 and Maraging steel. To obtain reference values to analyze the evolution of residual stresses, it was first assumed that the gear and the reconstructed tooth were made of the same material. Subsequently, it was considered that the gear was made of 42CrMo4 and the tooth was made of IN625 or Maraging steel, as shown in Figure 15.

Figure 34 and Figure 35 show the values obtained for the use of different materials. With the use of different materials, it was expected that there would be a discontinuity of properties and so the residual stresses would be higher. Note that the residual stresses in a gear are always concentrated at the root of the tooth, so with the use of different materials, the stresses should be even higher.

Analyzing the values in Figure 34, relative to the residual stresses induced by the DED process, it can be seen that the stresses increased when different materials were used. However, this increase when using 42CrMo4 steel in the gear base was not very significant.

The results obtained for spatial displacement, Figure 35, show a significant reduction when using 42CrMo4 steel. This reduction was more accentuated when Maraging steel was used, this may have been related to the different thermal expansions of the materials.

One of the causes of this difference is that Maraging steel is more sensitive to temperature variation than 42CrMo4 steel, as can be seen in the properties in Table 5 and Table 7.

In Figure 36 and Figure 37, it is possible to see the images obtained from the simulations carried out to study the influence of different materials on the residual stresses and spatial displacement. In these figures, In625 and 42CrMo4 steels were used.

In Figure 36, it can be seen that the use of different materials increased the residual stresses at the root of the tooth. This increase can be seen in the increase in the reddish spot.

On other hand, the spatial displacement decreased, as can be seen in Figure 37.

The results obtained with maraging and 42crmo4 steels, Figure 38 and Figure 39, were similar to those obtained with IN625.

## 5. Fatigue Analysis

In this section, the influence of fatigue on the gear obtained through the DED process will be studied. The DED process, used to reconstruct the gear, induces residual stresses in the part. To assess the impact of these stresses when conducting a fatigue test, a comparison was made with a gear without residual stresses.

### 5.1. Introduction

Fatigue has become increasingly crucial for technological advancement in automobiles, aircraft, compressors, turbines, and other systems subjected to repeated loading and vibration. Today, it is commonly stated that fatigue accounts for at least 90% of all mechanical service failures [50]. Fatigue failure is especially dangerous, because it occurs without warning. As a result, fatigue failure prediction methodologies are fundamental, and the S–N curve is one of the oldest stress-based methods for predicting fatigue failure [51].

The fatigue problems of gears, one of the most widely used mechanical elements in transmission systems, limit the reliability and longevity of machines, especially the typical high-cycle-fatigue machines, such as wind turbines and high-speed trains. Gear fatigue represents a gradual deterioration process of material properties and the continuous accumulation of damage. Cracks induced by damage accumulation eventually lead to final failures, such as pitting, spalling, or tooth failure [52].

Martukanitz and Simpson [53] reported that build rate and feature definition are closely linked and related to surface quality. Figure 40 illustrates the relationship between build rate, power, and feature quality. In general, as build rate increases, feature quality/resolution decreases. The implication is that for fatigue critical parts fabricated using high-deposition-rate AM processes, post process surface finishing may be necessary.

Fatigue loading with constant amplitude (or constant amplitude loading) is where all load cycles are identical. The Basquin relation can describe the stress–life relationship under high-cycle (or stress-controlled) fatigue conditions, where the significant controlling parameter is the elastic strain or stress level [54]: (3)$$\sigma_{a}=\frac{\Delta \sigma }{2}=\left(\sigma_{f}^{{}^{\prime }}-\sigma_{m}\right){\left(2N_{f}\right)}^{b}$$
(4)$$\Delta \sigma =(\sigma_{max}-\sigma_{min})$$
(5)$$\sigma_{m}=\frac{\sigma_{max}+\sigma_{min}}{2}$$
where $\sigma_{a}$ is the alternating stress amplitude, Δσ is the stress range, $\sigma_{f}^{{}^{\prime }}$ is the fatigue strength coefficient, $\sigma_{m}$ is the mean stress, *b* is the fatigue strength exponent, and $2N_{f}$ is the reversal to failure. The $\sigma_{max}$ and $\sigma_{min}$ correspond to the maximum and minimum stresses resulting from the stress cycle: applying and removing the load.

The fatigue strength coefficient $\sigma_{f}^{{}^{\prime }}$ and fatigue strength exponent *b*, for Inconel 625 were obtained for the S–N curve and are shown in the following relation [54]: (6)$$\sigma_{a}=\left(2282-\sigma_{m}\right){\left(2N_{f}\right)}^{-0.134}$$

### 5.2. Modeling Description

To perform stress simulation of the gear, a force was applied, as shown in Figure 41. The applied force was calculated from a moment of 215 Nm, the value to which this type of gear is usually subjected. The stress simulation was applied after the DED repair simulation, to analyze the influence of these residual stresses. Another stress simulation was performed, to enable comparison, and in this new simulation, it was assumed that the gear had no residual stresses resulting from the production process.

### 5.3. Results

The values corresponding to the range, mean, amplitude of stress, and the number of cycles to failure for the Mises and principal stresses are grouped in Table 18.

A comparison of the maximum stress must always consider the same node. However, the node where the stress is maximum at the end of a simulation does not correspond to the node where the stress is maximum at the beginning. Therefore, two distinct nodes were considered: one corresponding to the maximum stress at the beginning, and the other to the maximum stress at the end of the simulation. A fatigue simulation of a gear without previous residual stresses was also carried out; the results are also shown in Table 18.

Figure 42 represents the results presented in Table 18, which shows the number of cycles for the failure considering the Mises and the principal stresses. The simulation results in which there were no previous residual stresses (control) are also represented in this figure.

Figure 43 represents the nodes where the stresses are maximum; for the Mises stresses and the principal stresses, these values correspond to the end of the simulation.

## 6. Conclusions

The DED and PBF processes are the predominant technologies used worldwide, for that reason it is important to study the mechanical behavior and structural integrity of parts obtained using these two processes.

Regarding the NIST bridge benchmark, the objective initially set was achieved, since the results obtained in the simulations were similar to the results measured experimentally. The case of the benchmark bridge was also a way of verifying and validating the work developed.

This parametric study of the use of the DED process in gear repair was motivated by the GEAR3D project. Despite the many parameters existing in the DED process, only five were analyzed, thus representing a limitation of this work. The choice of mesh for the remaining simulations required a compromise between the computational cost and the accuracy of the results obtained. Although the scanning strategies chosen only differed in the deposition track angle, it was possible to conclude that the choice of an appropriate scanning strategy allowed reducing the residual stresses and therefore optimized the part manufacturing process.

This parametric study also concluded that the preheating temperature is a parameter that strongly influences the values of residual stresses induced in a manufactured part, with variations that can reach 500%. Regarding the study of the time step influence, there were no significant differences in the results obtained, mainly in the residual stresses. These results did not correspond well to what was expected, so a more detailed analysis could be carried out to understand the progressive activation of the material and its influence on the results.

Finally, taking into account the results obtained for the influence of the use of different materials, it was possible to verify that the residual stresses actually changed with the use of different materials. On the other hand, the spatial displacement decreased substantially. One of the reasons for this may have been related to the different expansion coefficients of the materials.

The fatigue analysis concluded that when there are residual stresses from the AM process, the stresses have low amplitudes and high mean values, which can result in a longer life than without residual stresses, where there are higher ranges and lower average stresses. Thus, high residual stresses can lead to a redistribution of elastoplastic stresses, which turns out to be beneficial in terms of fatigue.

However, there is still much work to be done in this field. Future research should analyze other parameters that influence the metal additive manufacturing process. Future work should also focus on validating the simulations performed using on-gear reparation.

In conclusion, AM processes are complex, due to the number of parameters that can be changed and their impact on the obtained parts. This work significantly contributes to the knowledge of the influence of some of the parameters of the DED process. In addition, studying the fatigue of gears produced is an essential outcome in the perception of the influence of residual stresses induced during the gear repair process.

## Figures and Tables

**Figure 1 materials-16-03549-f001:**
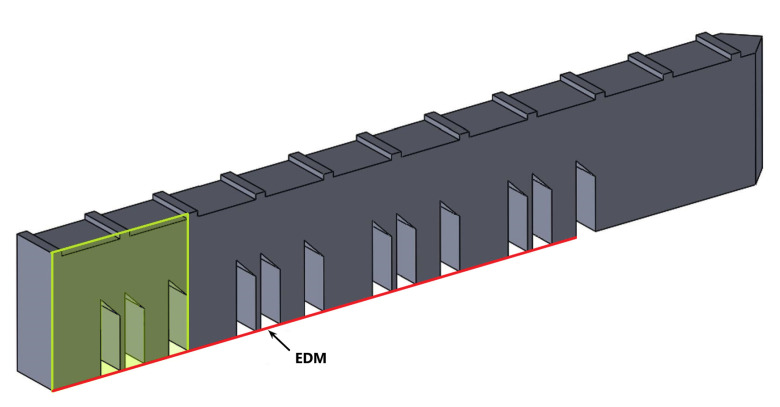
Benchmark bridge structure and EDM cut location (red line). The green box is the pattern that repeats four times.

**Figure 2 materials-16-03549-f002:**
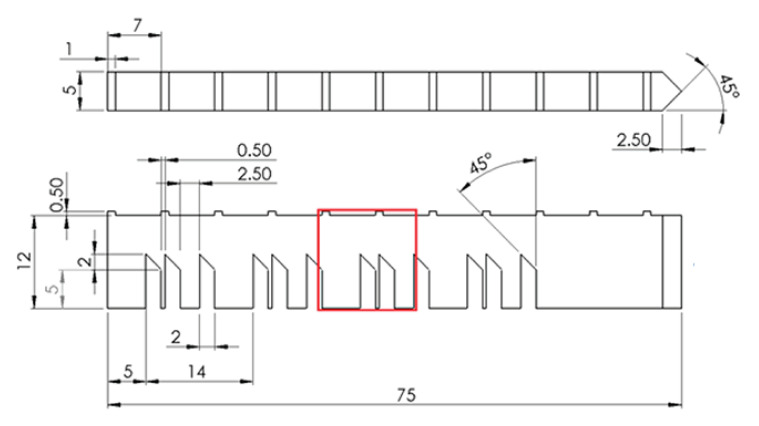
Bridge dimensions (in millimeters). The red box represents where the in-situ measurements were taken [23].

**Figure 3 materials-16-03549-f003:**
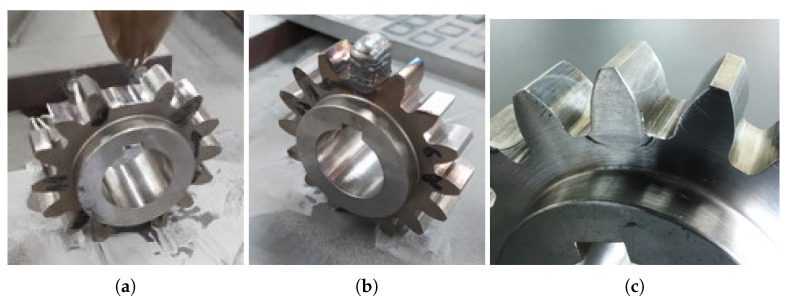
Gear to repair: (**a**) gear before repair; (**b**) gear after repair; (**c**) machined gear [27].

**Figure 4 materials-16-03549-f004:**
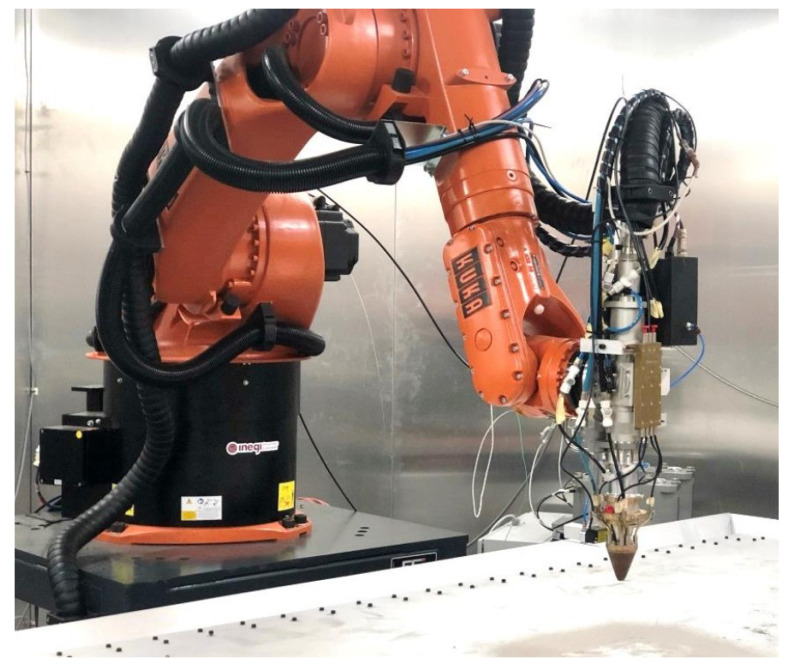
INEGI’s DED system setup [27].

**Figure 5 materials-16-03549-f005:**
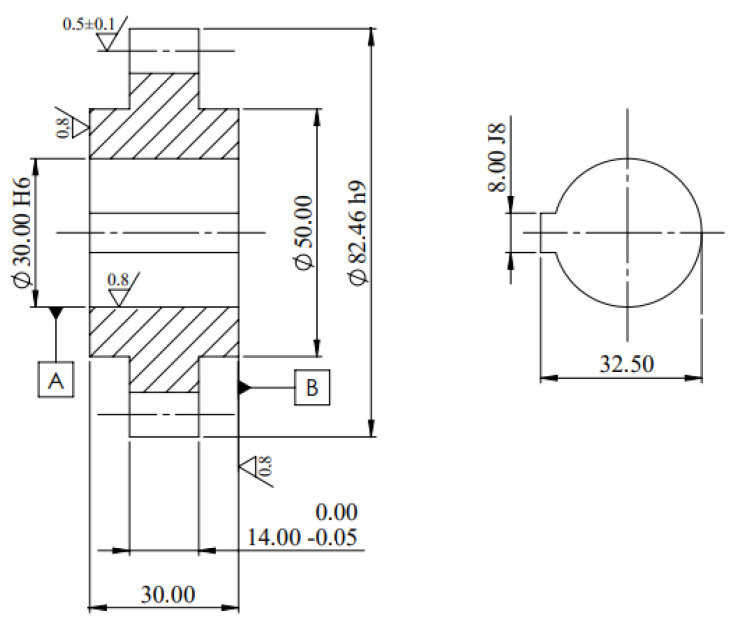
Gear dimensions (millimeters) [27].

**Figure 6 materials-16-03549-f006:**
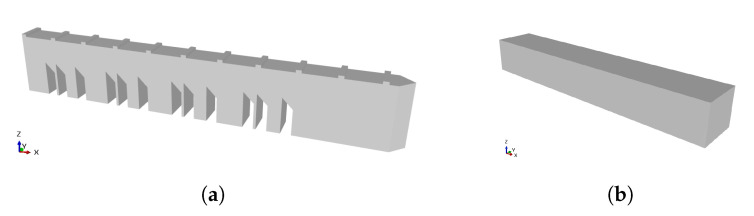
Benchmark bridge (AMB2018-01): (**a**) bridge, (**b**) build plate.

**Figure 7 materials-16-03549-f007:**
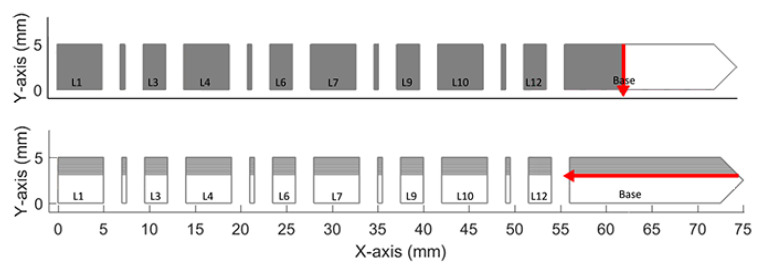
Description of the scan pattern strategy [23].

**Figure 8 materials-16-03549-f008:**
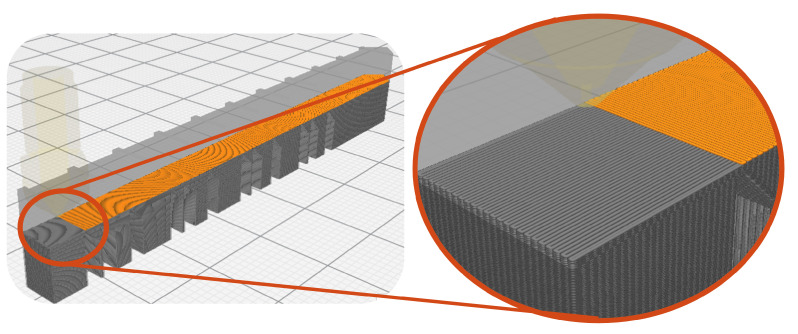
Scan strategy—bridge benchmark.

**Figure 9 materials-16-03549-f009:**
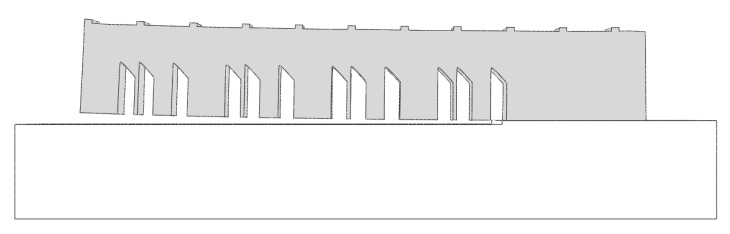
Nist benchmark bridge—upward deflection.

**Figure 10 materials-16-03549-f010:**
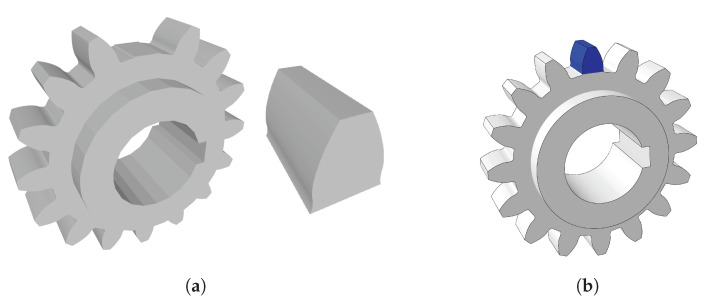
Gear Modeling—(**a**) gear and tooth belonging to the two-part model, (**b**) one-part model.

**Figure 11 materials-16-03549-f011:**
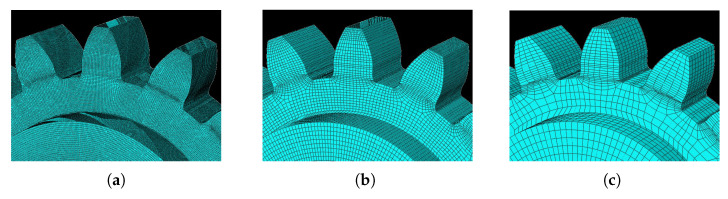
Mesh variation: (**a**) Mesh-1 size of 0.25 mm, (**b**) Mesh-2 size of 0.50 mm, (**c**) Mesh-3 size of 1.50 mm.

**Figure 12 materials-16-03549-f012:**
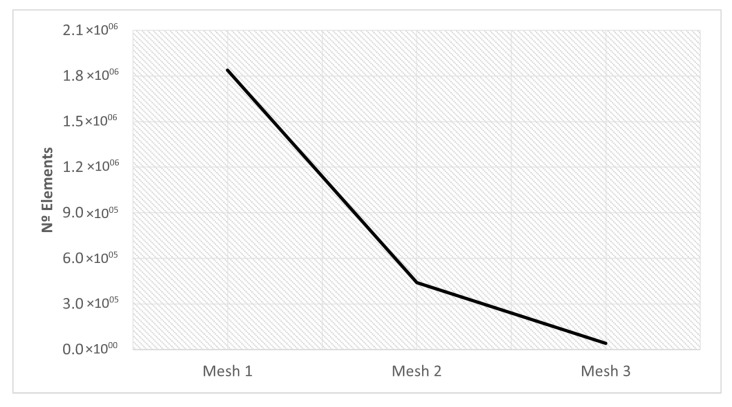
Mesh parameters— number of elements and mesh size.

**Figure 13 materials-16-03549-f013:**
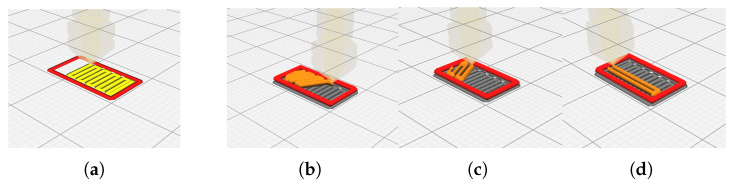
Scanning strategy I: (**a**) base layer 90°, (**b**) scan-1 120°, (**c**) scan-2 60°, (**d**) scan-3 0°.

**Figure 14 materials-16-03549-f014:**
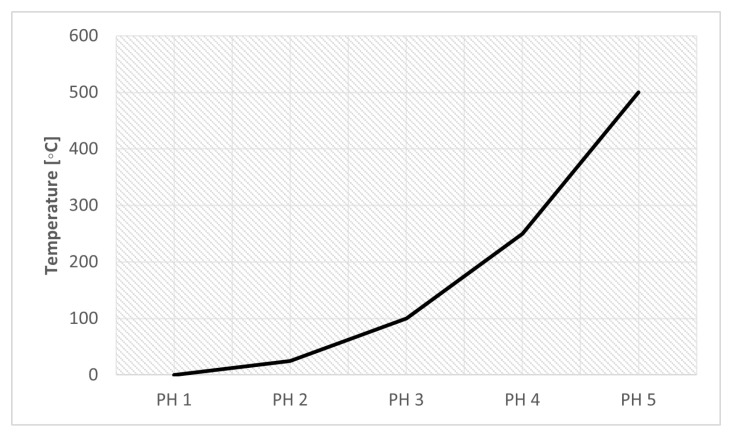
Preheat Temperature Influence.

**Figure 15 materials-16-03549-f015:**
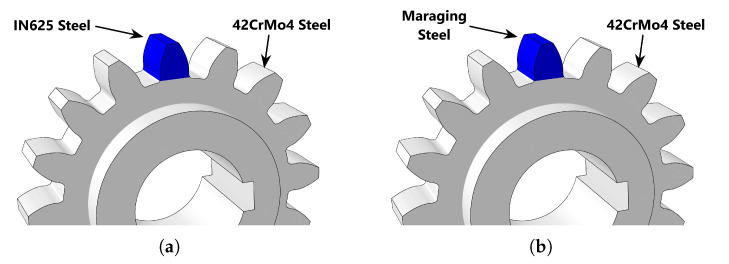
Gear repaired with different materials: (**a**) IN625 + 42CrMo4 steel. (**b**) Maraging steel + 42CrMo4 steel.

**Figure 16 materials-16-03549-f016:**
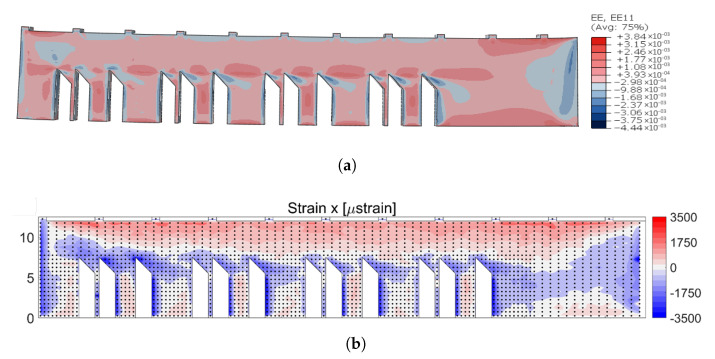
Comparison of elastic strain results in the x-direction: (**a**) simulation result EE11 (from $-4.44\times 10^{-3}$ to $3.84\times 10^{-3}$), (**b**) strain measurements from NIST [23].

**Figure 17 materials-16-03549-f017:**
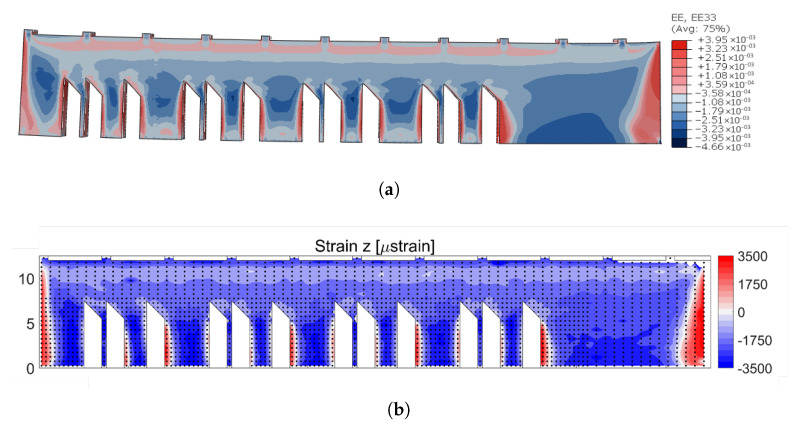
Comparison of the elastic strain results in the x-direction: (**a**) Simulation result EE33, (**b**) strain measurements from NIST [23].

**Figure 18 materials-16-03549-f018:**
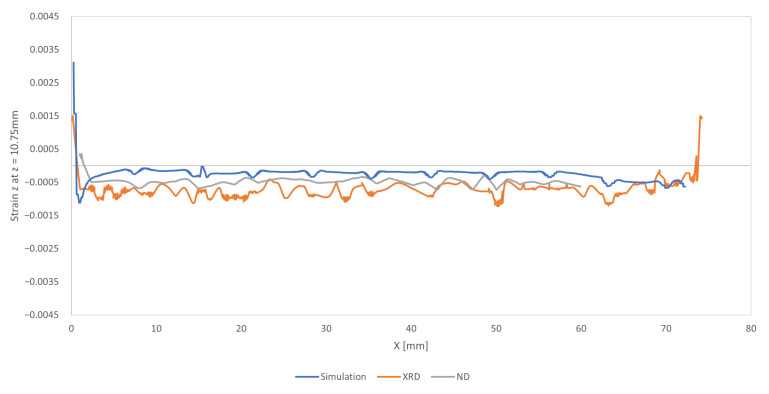
Benchmark bridge—simulation result for elastic strain z at z = 10.75 mm.

**Figure 19 materials-16-03549-f019:**
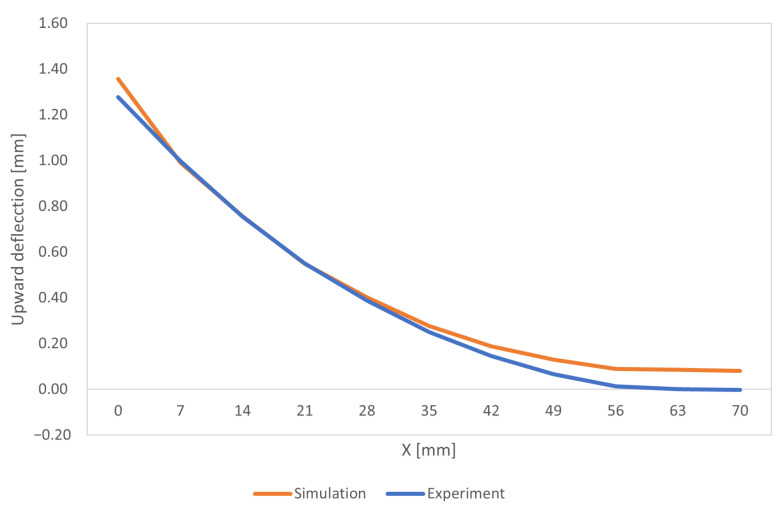
Benchmark bridge—upward deflection.

**Figure 20 materials-16-03549-f020:**
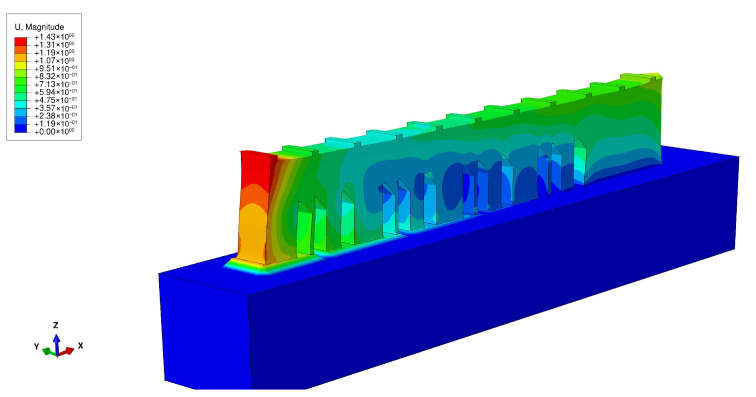
NIST benchmark bridge—deflection.

**Figure 21 materials-16-03549-f021:**
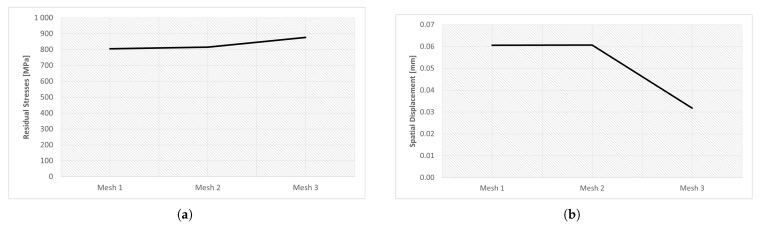
Results of mesh size study: (**a**) residual stresses, (**b**) spatial displacement.

**Figure 22 materials-16-03549-f022:**
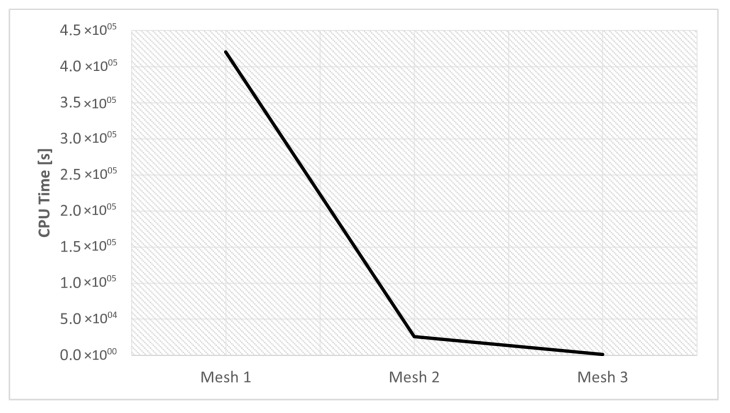
CPU time mesh simulations.

**Figure 23 materials-16-03549-f023:**
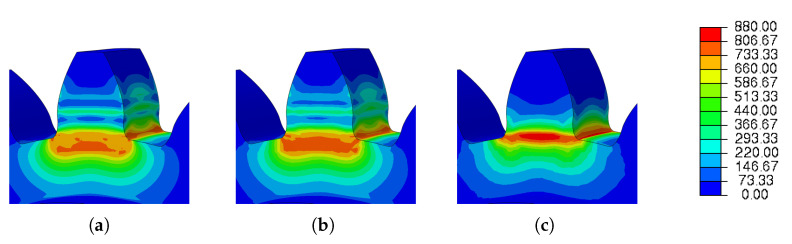
Results of mesh variation—residual stress: (**a**) Mesh 1; (**b**) Mesh 2 and (**c**) Mesh 3.

**Figure 24 materials-16-03549-f024:**
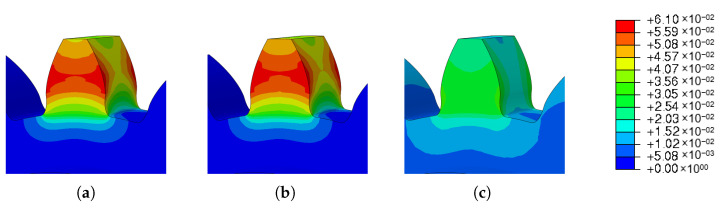
Results of mesh variation—spatial displacement: (**a**) Mesh 1; (**b**) Mesh 2 and (**c**) Mesh 3.

**Figure 25 materials-16-03549-f025:**
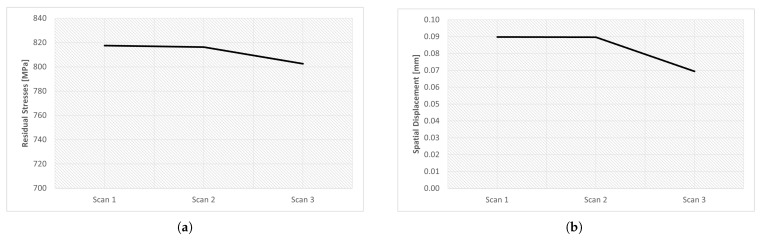
Results of the parametric study of the scan variation: (**a**) residual stresses (**b**) spatial displacement.

**Figure 26 materials-16-03549-f026:**
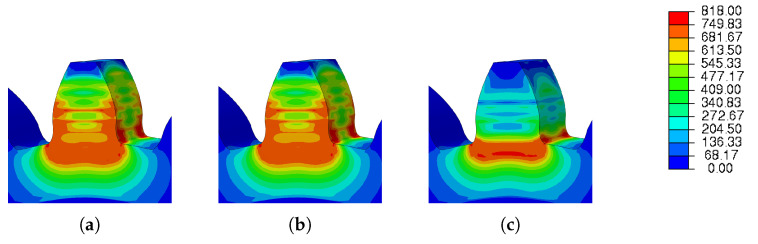
Scan strategy—residual Stress: (**a**) Scan 1, (**b**) Scan 2, (**c**) Scan 3.

**Figure 27 materials-16-03549-f027:**
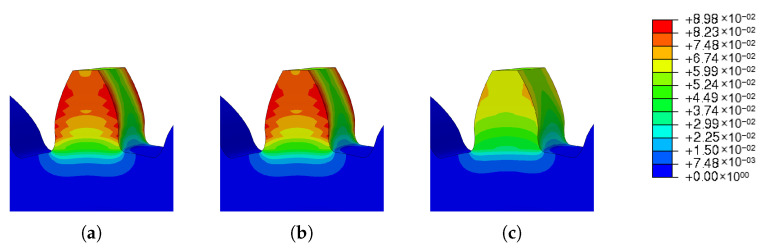
Scan strategy—Spatial displacement: (**a**) Scan 1, (**b**) Scan 2, (**c**) Scan 3.

**Figure 28 materials-16-03549-f028:**
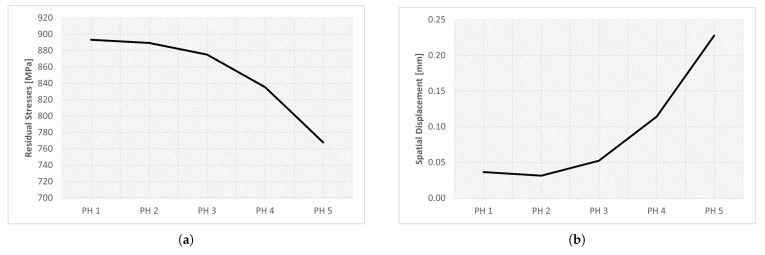
Results of the parametric study of the preheat temperature variation: (**a**) residual stresses, (**b**) spatial displacement.

**Figure 29 materials-16-03549-f029:**
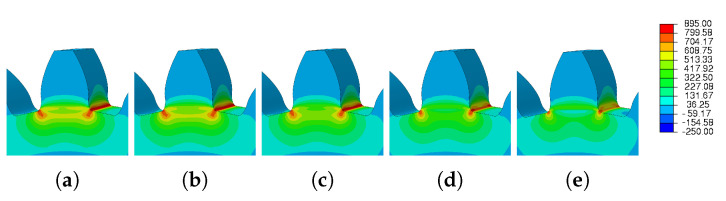
Preheat temperature—residual stress: (**a**) Ph-1, (**b**) Ph-2, (**c**) Ph-3, (**d**) Ph-4, (**e**) Ph-5.

**Figure 30 materials-16-03549-f030:**
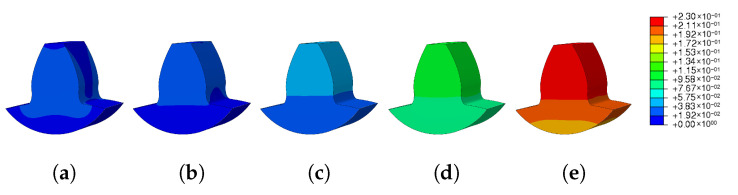
Preheat temperature—spatial displacement: (**a**) Ph-1, (**b**) Ph-2, (**c**) Ph-3, (**d**) Ph-4, (**e**) Ph-5.

**Figure 31 materials-16-03549-f031:**
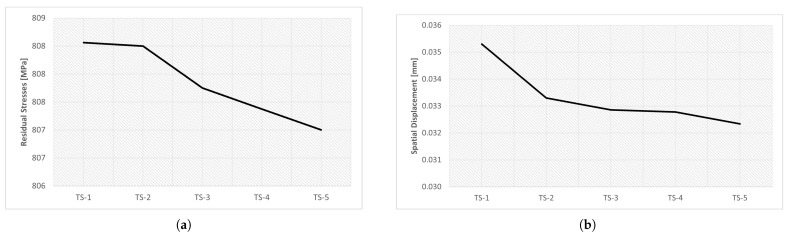
Results of the parametric study of the step time variation: (**a**) residual stresses, (**b**) spatial displacement.

**Figure 32 materials-16-03549-f032:**
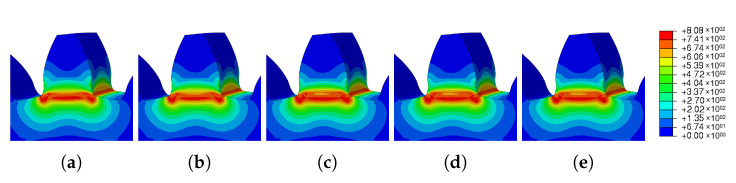
Influence of time step—residual stress: (**a**) TS-1 (**b**) TS-2 (**c**) TS-3 (**d**) TS-4 (**e**) TS-5.

**Figure 33 materials-16-03549-f033:**
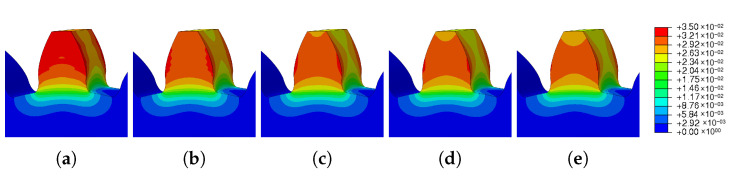
Influence of time step—spatial displacement: (**a**) TS-1, (**b**) TS-2, (**c**) TS-3, (**d**) TS-4, (**e**) TS-5.

**Figure 34 materials-16-03549-f034:**
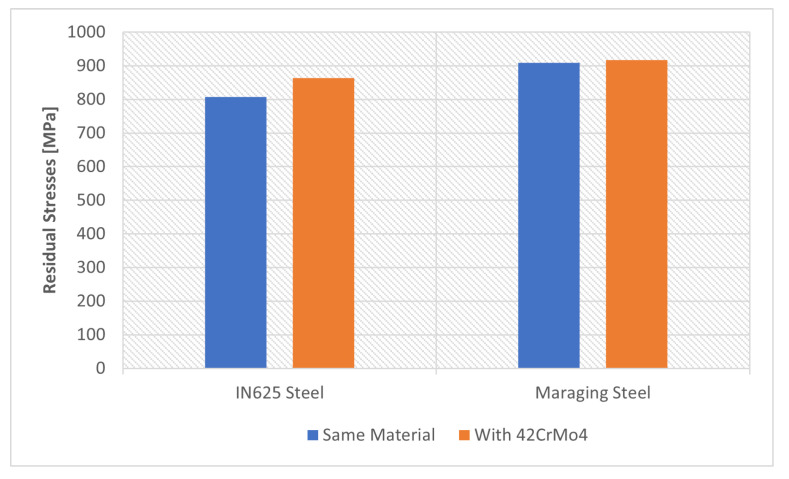
Material influence on residual stress.

**Figure 35 materials-16-03549-f035:**
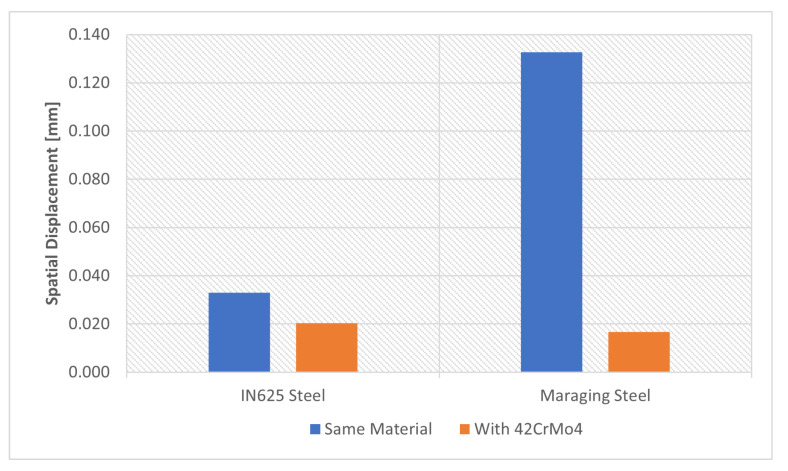
Material influence on spatial displacement.

**Figure 36 materials-16-03549-f036:**
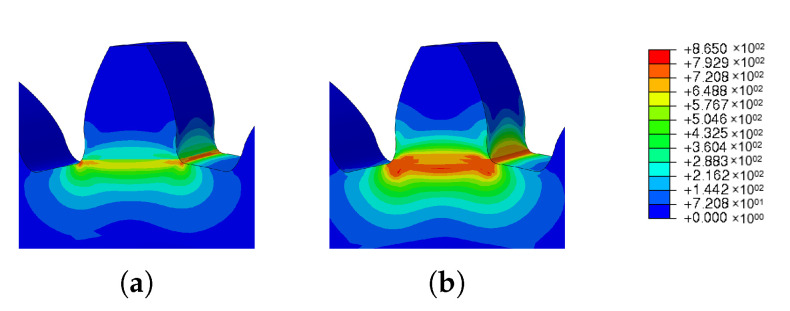
Material influence on residual stresses simulation results: (**a**) IN625, (**b**) IN625 with 42CrMo4 steel.

**Figure 37 materials-16-03549-f037:**
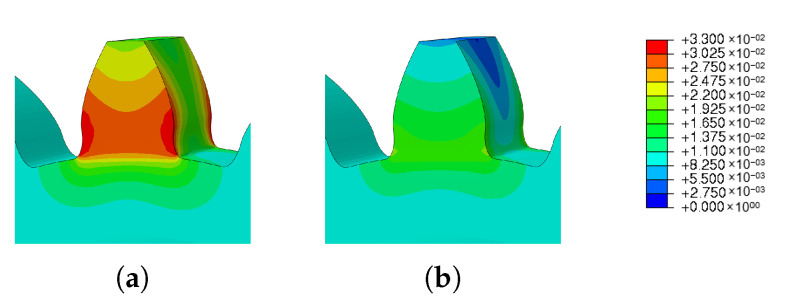
Material influence on spatial displacement simulation results: (**a**) IN625, (**b**) IN625 with 42CrMo4 steel.

**Figure 38 materials-16-03549-f038:**
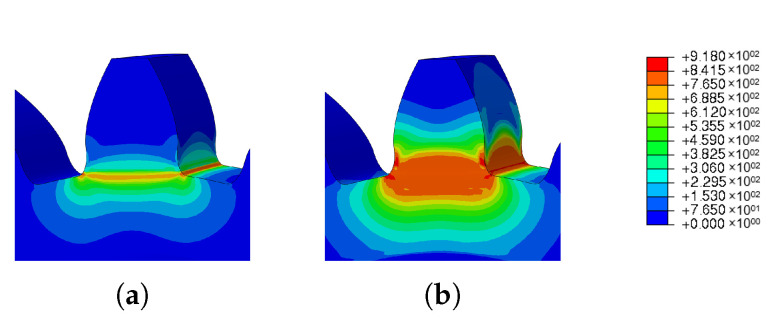
Material influence on residual stresses simulation results: (**a**) maraging steel, (**b**) maraging steel with 42CrMo4 steel.

**Figure 39 materials-16-03549-f039:**
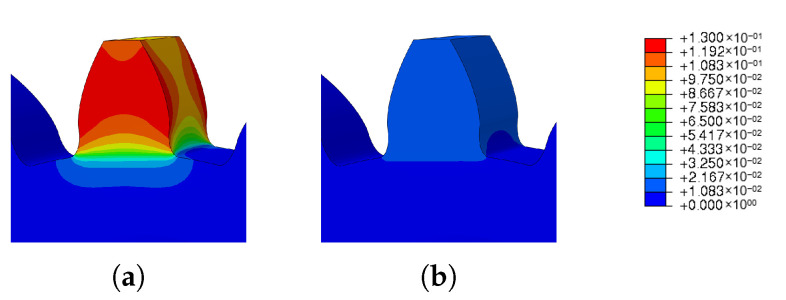
Material influence on spatial displacement simulation results: (**a**) maraging steel, (**b**) maraging with 42CrMo4 steel.

**Figure 40 materials-16-03549-f040:**
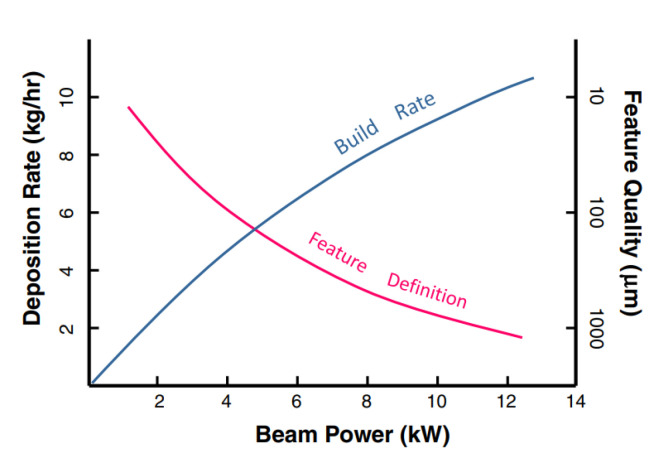
Relationship between build rate, power, and feature definition [53].

**Figure 41 materials-16-03549-f041:**
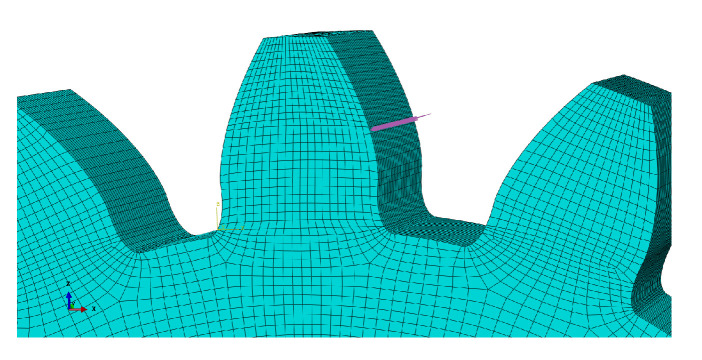
Force application in the simulation model of the repaired gear.

**Figure 42 materials-16-03549-f042:**
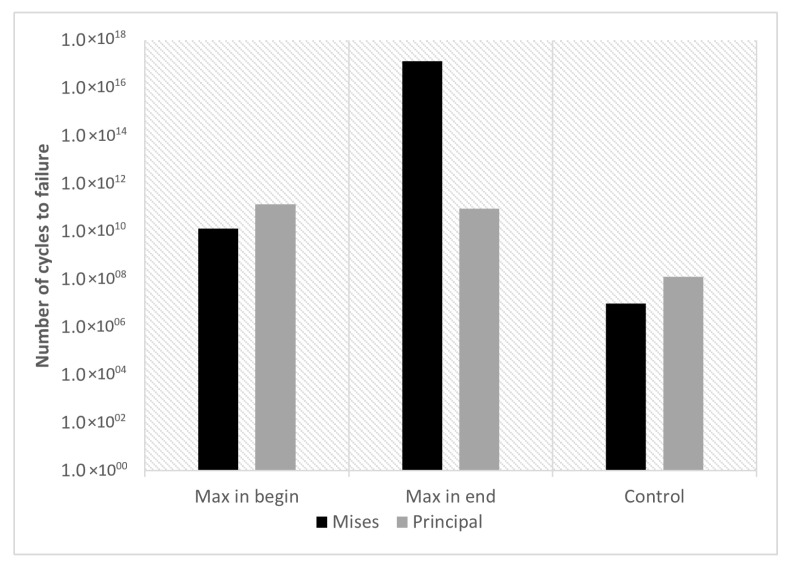
Results from the stress simulation—number of cycles to failure.

**Figure 43 materials-16-03549-f043:**
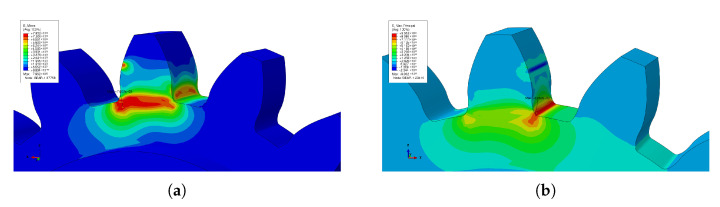
Stress simulation results (end of simulation): (**a**) max Mises stress, (**b**) max princiapl stress.

**Table 1 materials-16-03549-t001:** Parameters used in the DED process of gear simulation.

Layer Height (mm)	Line Width (mm)	Laser Power (w)	Scan Speed (mm/s)	Hatch Space (mm)
1.1	1.0	1800	111	1.0

**Table 2 materials-16-03549-t002:** Latent heat properties [28].

Latent Heat of Fusion (mJ/ton)	Solidus Temperature (°C)	Liquidus Temperature (°C)
$272\times 10^{9}$	1290	1350

**Table 3 materials-16-03549-t003:** Thermo-physical properties of IN625 [28].

Temperature (°C)	Specific Heat (mJ/ton °C)	Conductivity (mW/mm °C)	Thermal Expansion (1/°C)
21	4.10 $\times 10^{8}$	9.8	
93	4.27 $\times 10^{8}$	10.8	1.28 $\times 10^{-5}$
204	4.56 $\times 10^{8}$	12.5	1.31 $\times 10^{-5}$
316	4.81 $\times 10^{8}$	14.1	1.33 $\times 10^{-5}$
427	5.11 $\times 10^{8}$	15.7	1.37 $\times 10^{-5}$
538	5.36 $\times 10^{8}$	17.5	1.40 $\times 10^{-5}$
649	5.65 $\times 10^{8}$	19.0	1.48 $\times 10^{-5}$
760	5.90 $\times 10^{8}$	20.8	1.53 $\times 10^{-5}$
871	6.20 $\times 10^{8}$	22.8	1.58 $\times 10^{-5}$
982	6.45 $\times 10^{8}$	25.2	
1093	6.70 $\times 10^{8}$		

**Table 4 materials-16-03549-t004:** Young’s modulus of IN625 [28].

Temperature (°C)	Young’s Modulus (MPa)
21	2.075 $\times 10^{5}$
93	2.041 $\times 10^{5}$
204	1.979 $\times 10^{5}$
316	1.917 $\times 10^{5}$
427	1.855 $\times 10^{5}$
538	1.786 $\times 10^{5}$
649	1.703 $\times 10^{5}$
760	1.606 $\times 10^{5}$
871	1.475 $\times 10^{5}$

**Table 5 materials-16-03549-t005:** Thermophysical properties of maraging steel [32].

Temperature (°C)	Specific Heat (mJ/ton·°C)	Conductivity (mW/mm·°C)	Thermal Expansion (1/°C)
20	4.449 $\times 10^{8}$	15.81	1.8370 $\times 10^{-6}$
100	4.747 $\times 10^{8}$	17.46	1.0797 $\times 10^{-5}$
200	5.121 $\times 10^{8}$	19.52	2.1997 $\times 10^{-5}$
300	5.495 $\times 10^{8}$	21.58	3.3197 $\times 10^{-5}$
400	5.869 $\times 10^{8}$	23.64	4.4397 $\times 10^{-5}$
500	6.243 $\times 10^{8}$	25.70	5.5597 $\times 10^{-5}$
600	6.617 $\times 10^{8}$	27.76	6.6797 $\times 10^{-5}$
700	6.991 $\times 10^{8}$	29.82	7.7997 $\times 10^{-5}$
800	7.365 $\times 10^{8}$	31.88	8.9197 $\times 10^{-5}$
900	7.739 $\times 10^{8}$	33.94	1.00397 $\times 10^{-4}$
1000	8.113 $\times 10^{8}$	36.00	1.11597 $\times 10^{-4}$

**Table 6 materials-16-03549-t006:** Johnson–Cook plasticity coefficients for maraging steel 300 [20].

*A* (MPa)	*B* (MPa)	*n*	*m*	*C*	${\dot{\varepsilon }}_{0}$
758.423	172.147	0.2258	0.7799	0.0522	70

**Table 7 materials-16-03549-t007:** Thermophysical properties of 42CrMo4 steel [34].

Temperature (°C)	Specific Heat (mJ/ton°C)	Conductivity (mW/mm°C)	Thermal Expansion (1/°C)
20	4.60 $\times 10^{8}$		
100	4.80 $\times 10^{8}$	42.7	1.22 $\times 10^{-5}$
200	4.73 $\times 10^{8}$	42.3	1.26 $\times 10^{-5}$
400	5.19 $\times 10^{8}$	37.7	1.36 $\times 10^{-5}$
600	5.61 $\times 10^{8}$	33.1	1.45 $\times 10^{-5}$

**Table 8 materials-16-03549-t008:** Johnson–Cook Plasticity coefficients for 42CrMo4 steel.

*A* (MPa)	*B* (MPa)	*n*	*m*	*C*	${\dot{\varepsilon }}_{0}$
595	580	0.133	1.03	0.023	70
852	1102	0.008	0.7	1	1000

**Table 9 materials-16-03549-t009:** 42CrMo4 steel’s latent heat properties.

Latent Heat of Fusion (mJ/ton)	Solidus Temperature (°C)	Liquidus Temperature (°C)
250 $\times 10^{9}$	1399	1410

**Table 10 materials-16-03549-t010:** Parameters used in the AM process of the benchmark bridge.

Layer Thickness	Laser Power	Scan Speed	Hatch Space
0.02 mm	195 w	900 mm/s	0.1 mm

**Table 11 materials-16-03549-t011:** Mesh parameters—number of elements and mesh size.

	Number of Elements	Mesh Size (mm)
Mesh 1	1,838,455	0.25
Mesh 2	439,768	0.50
Mesh 3	41,832	1.50

**Table 12 materials-16-03549-t012:** Influence of the preheat temperature.

Name	Ph-1	Ph-2	Ph-3	Ph-4	Ph-5
**Temperature [°C]**	0	25	100	250	500

**Table 13 materials-16-03549-t013:** Influence of the Time Step.

Name	Initial Increment	Max. Increment
TS-1	5.0	15.0
TS-2	1.0	6.0
TS-3	0.5	3.0
TS-4	0.1	1.5
TS-5	0.05	0.3

**Table 14 materials-16-03549-t014:** Mesh results for the gear case study.

	Spatial Displacement (mm)	Residual Stress (MPa)	Number Elements	Mesh Size (mm)	CPU Time (s)
Mesh 1	$6.058\times 10^{-02}$	804.4	1,838,455	0.25	420,520.0
Mesh 2	$6.072\times 10^{-02}$	815.8	439,768	0.50	25,716.0
Mesh 3	$3.182\times 10^{-02}$	876.2	41,832	1.50	1227.7

**Table 15 materials-16-03549-t015:** Scan Strategy Results.

	Spatial Displacement (mm)	Residual Stress (MPa)	Angle (°)
Scan 1	$8.98\times 10^{-02}$	817.5	[90,120]
Scan 2	$8.97\times 10^{-02}$	816.2	[90,60]
Scan 3	$6.94\times 10^{-02}$	802.5	[90,0]

**Table 16 materials-16-03549-t016:** Preheat Results.

	Spatial Displacement (mm)	Residual Stress (MPa)	Temperature (°C)
Ph-1	$3.63\times 10^{-02}$	893.2	0
Ph-2	$3.15\times 10^{-02}$	889.3	25
Ph-3	$5.25\times 10^{-02}$	875.0	100
Ph-4	$1.14\times 10^{-01}$	835.4	250
Ph-5	$2.28\times 10^{-01}$	767.8	500

**Table 17 materials-16-03549-t017:** Results of step time variation.

	Spatial Displacement (mm)	Residual Stresses (MPa)	Initial Increment	Max. Increment
TS-1	$3.53\times 10^{-02}$	808.45	5.00	15.0
TS-2	$3.33\times 10^{-02}$	808.40	1.00	6.0
TS-3	$3.29\times 10^{-02}$	807.80	0.50	3.0
TS-4	$3.28\times 10^{-02}$	807.50	0.10	1.5
TS-5	$3.23\times 10^{-02}$	807.20	0.05	0.3

**Table 18 materials-16-03549-t018:** Results of the stress simulation.

Mises Stress				
	** Δσ ** **(MPa)**	** $\sigma_{m}$ ** **(MPa)**	** $\sigma_{a}$ ** **(MPa)**	$N_{f}$
Max in begin	122.54	752.05	61.27	$1.34\times 10^{10}$
Max in end	13.92	779.24	6.96	$1.32\times 10^{17}$
**Principal Stress**				
	** Δσ ** **(MPa)**	** $\sigma_{m}$ ** **(MPa)**	** $\sigma_{a}$ ** **(MPa)**	$N_{f}$
Max in begin	83.32	861.93	41.66	$1.37\times 10^{11}$
Max in end	87.66	862.37	43.83	$9.35\times 10^{10}$
**Control (no previous residual stresses)**				
	** Δσ ** **(MPa)**	** $\sigma_{m}$ ** **(MPa)**	** $\sigma_{a}$ ** **(MPa)**	$N_{f}$
Mises Stress	434.79	217.39	217.39	$9.87\times 10^{06}$
Principal Stress	316.92	158.46	158.46	$1.29\times 10^{08}$

## Data Availability

The data that support the findings of this study are available from the corresponding author upon reasonable request.

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
