# Peer review of "Direct Energy Deposition Parametric Simulation Investigation in Gear Repair Applications"

_materials, 2023, doi:10.3390/ma16093549_

Round 1
Reviewer 1 Report
The study conducted aims to predict the tensions and distortions imposed in the gear repair process by directed energy deposition. The study is well organized and somewhat innovative, but still needs to be explained in the following areas:
(1) The introduction of the experimental equipment is not sufficient. The parameters and pictures of related equipment and instruments need to be supplemented.
(2) How are the results of the experiments measured using the sensors? Can refer to: 10.1016/j.ymssp.2021.108386 and 10.1109/TAES.2023.3257777.
(3) Why is the finite element model in Figure 10 not considered to be cyclically symmetrical? Because discretization of all geometric models does not guarantee a completely consistent mesh shape for each sub-model.
(4) Is the thermal-mechanical coupling study in this manuscript only simulation?
(5) The latest research progress on the temperature field needs to be supplemented: 10.1016/j.ast.2023.108155.
(6) F10 and F40 do not use the same set of finite element meshes, why?
Reviewer 2 Report
Enclosed as a separate file.

Round 2
Reviewer 1 Report
good job
Reviewer 2 Report
Now the manuscript can be accepted for the publication.